# Pharmacist-Led Interventions for Medication Adherence in Patients with Chronic Kidney Disease: A Scoping Review

**DOI:** 10.3390/pharmacy11060185

**Published:** 2023-11-30

**Authors:** Luke Calleja, Beverley D. Glass, Alice Cairns, Selina Taylor

**Affiliations:** 1Pharmacy, College of Medicine and Dentistry, James Cook University, Townsville 4811, Australia; 2Australian Institute of Tropical Health and Medicine, James Cook University, Cairns 4870, Australia; 3Murtupuni Mount Isa Centre for Rural and Remote Health, Mount Isa 4825, Australia

**Keywords:** barriers, interventions, patient perceptions, renal, compliance, medicines

## Abstract

Background: Patients with chronic kidney disease (CKD) are routinely prescribed complex medication regimes. Medication reconciliation, medicine reviews, patient counselling and disease state and medication education are all key pharmacist-led interventions, which can improve medication adherence in patients with CKD. Aim: To characterize peer reviewed literature on the role of pharmacists in supporting medication adherence of patients with chronic kidney disease and highlight the impact they might have in the health outcomes for patients. Method: This review was performed in accordance with the Scoping Review Framework outlined in the Joanna Briggs Institute Reviewer’s Guide. Four electronic databases were searched (Medline (Ovid), Emcare, Scopus and Web of Science) for all relevant literature published up until November 2022. A total of 32 studies were reviewed against an exclusion and inclusion criteria, with findings from each study categorized into barriers, interventions, perceptions, financial implications and outcomes. Results: Eight eligible studies were identified, where pharmacists’ interventions including medication reconciliation, medicine reviews, patient counselling and disease state and medication education, were all reported to have a positive effect on medication adherence. Although pharmacy services in chronic kidney disease were acceptable to patients and pharmacists, these services were under-utilized and limited by logistical constraints, including staffing shortages and time limitations. Patient education supplemented with education tools describing disease states and medications was reported to increase patient adherence to medication regimes. Conclusions: Pharmacist-led interventions play an integral role in improving medication adherence in patients with chronic kidney disease, with their inclusion in renal care settings having the potential to improve outcomes for patients.

## 1. Introduction

Chronic kidney disease (CKD) is characterized by functional and structural changes to the renal system, most commonly secondary to vascular complications of diabetes mellitus, hypertensive organ-end damage to renal vasculature or glomerulonephritis, which lasts longer than three months due to long-term exposure to toxins [1,2]. CKD places a large financial burden on the Australian healthcare system, costing approximately AUD 5.1 billion per annum, with 1.3 million CKD-related hospitalizations every year [3]. Rural and remote areas report almost twice as many people with CKD each year compared to major cities [3]. Aboriginal and Torres Strait Islander people with CKD experience the greatest burden of illness, with five times more presentations to hospital, which may be a result of remoteness and/or socioeconomic disadvantage [3]. Treatment of CKD and associated co-morbidities often requires therapy regimes with many different medications, to mitigate acute symptoms and halt or slow the progression of the renal impairment [1,2,4].

Numerous barriers to effective pharmacotherapy in patients with CKD exist, with nonadherence to prescribed medicines predominating [5]. Medication nonadherence (not taking medication as prescribed) leads to accelerated progression of renal failure and consequentially higher morbidity and mortality rates [1,2]. The World Health Organization (WHO) suggests that there are five factors which are determinants of patient medication adherence [6]. The donut model [6] stratifies these factors in order of importance: (1) patient-related factors, (2) socioeconomic factors, (3) healthcare system and healthcare professional factors, (4) medication-related factors and (5) condition-related factors [6]. All healthcare professionals, including physicians, nurses and pharmacists, play a role in identifying and mitigating any factors, which may limit a patient’s medication adherence at each stage of their healthcare journey [7]. However, this can become complex and challenging with patients who are on extensive medication regimes characteristic of CKD [1,2,8]. Therefore, healthcare interventions to support patient adherence to CKD-related medication regimes rarely occur regularly in practice [7]. Pharmacist-led interventions can include medication reviews, motivational counselling and hospital discharge care transition plans and may play an integral role in mitigating factors that negatively affect medication adherence [9]. However, the effective implementation of these interventions and the application of current adherence measurement practices in dialysis settings are not yet well established or investigated [9]. Patients with advanced CKD also commonly undergo hemodialysis therapy on a regular basis, which often presents as an opportunity for pharmacist medication interventions [9,10]. Although it has been reported that pharmacists have a role in clinical nephrology settings, this role remains unclear [11]. An improved understanding of the pharmacist’s role in supporting medication adherence in the CKD setting will allow greater consideration of pharmacist intervention effectiveness and best practices for this complex disease. 

### Aim

The aim of this scoping review was to characterize the literature reporting pharmacist-led interventions for improving medication adherence in patients with CKD to facilitate future research and systematic reviews examining the effect they might have in the health outcomes for patients with CKD.

## 2. Method

This scoping review followed the Scoping Review Framework outlined in the Joanna Briggs Institute Reviewer’s Manual which was originally proposed by Arksey and O’Malley and has since been developed and refined [12]. This framework provides a platform to describe evidence-based healthcare and effectively and rigorously synthesize evidence related to a topic area. The review followed the framework to: identify the research question; search for relevant studies; select studies, chart the data and collate and summarize the results. It provides systematic, transparent and trustworthy appraisal of resources for research synthesis within this review [12]. This approach allowed the incorporation of a range of study designs and addressed questions beyond those limited to intervention efficacy. 

### 2.1. Study Selection

The following outlines the ***inclusion criteria*** for the search strategy. 

#### 2.1.1. Participants

Studies conducted with patients with a diagnosis of CKD were included in this review. This included patients with all stages of chronic kidney disease (but excluded acute kidney failure). It also included those undergoing dialysis treatment and those who had received a transplant.

#### 2.1.2. Concept

The concept of a pharmacist-led or pharmacy-based service was applied to an intervention to improve medication adherence for patients with CKD.

#### 2.1.3. Context

Studies conducted in clinical setting with no restriction to region, country or geographical area were considered for this review.

#### 2.1.4. Types of sources

Studies with any type of research design were included.

### 2.2. Search Strategy

The key terms of this review were defined as ‘adherence’, ‘pharmacist’, ‘medicines’ and ‘chronic kidney disease’. These terms were inputted into Medline (Ovid) to develop an extensive list of applicable variations (e.g., pharmacist, pharmacy, pharmaceutics) to assist the search. Those keywords were combined with Boolean operators and applied to search the Medline (Ovid), Emcare, Scopus and Web of Science databases. The final search string was Pharmacist OR Pharmacists OR Pharmacy **AND** Medicine OR Therapy OR Drugs OR Tablets **AND** Renal Insufficiency OR Chronic Kidney Disease OR CKD OR Renal Disease OR Kidney Failure OR Kidney Disease **AND** Compliance OR Adherence. A date range of the previous 10 years (for recency of evidence) was applied to the search. All search terms were limited to title and abstract and the electronic search was supplemented by a manual search of the reference lists of the identified relevant studies (within 10 years of publication). The search was conducted in November 2022.

### 2.3. Extraction of Results and Data Synthesis

The search results were exported to the Endnote Version 9 [13] software to facilitate the screening process and duplicates were removed. Title, abstract and full text screening was conducted by study authors LC and BG based on the study selection, with all minor discrepancies between investigators resolved by discussion, until consensus was reached. Sources where the full text or English language articles could not be retrieved were excluded. The PRISMA extension for scoping review framework (PRISMA-ScR) was used to guide the search and data extraction [14]. Articles were included if they reported pharmacy or pharmacist supported management of medication adherence for patients with reported CKD or its major comorbidities. Data were extracted by study authors LC and BG using an extraction spreadsheet and were reviewed by ST and AC for verification, and consensus was reached by discussion with all investigators who collaborated via videoconference on multiple occasions over three months to discuss articles and interpret findings. The data charting process was extensive and included extracting, analyzing and presenting evidence from each study to answer the review question. Data items were initially predefined and were revised and refined through an iterative process with the study authors to form Table 1. The articles included in the review were then objectively analyzed, compared and synthesized. 

## 3. Results

### 3.1. Study Selection

Figure 1 shows the PRISMA extension for scoping review (PRISMA-ScR) flow diagram with the electronic database searching yielding 155 studies, and 30 studies were excluded as duplicate texts. Screening of the remaining 125 studies resulted in eight meeting the inclusion criteria. An additional 94 studies were excluded due to either not discussing the medication adherence in CKD or one of its major concomitant conditions or the role of the pharmacist in medication adherence. Sixteen articles were not within the date range. No additional studies were found through screening the reference lists. 

### 3.2. Characteristics of Included Studies

Table 1 summarizes each paper included in this review and reports findings that are subcategorized according to major themes: ‘Barriers’, ‘Interventions’, ‘Financial Implications’, ‘Patient Perceptions’ and overall ‘Outcomes’ of the study. Of the eight articles included, study designs included an online survey completed by pharmacists (*n* = 1) [9], a retrospective study of observational and descriptive data (*n* = 1) [15], prospective interventional trials (*n* = 5) [10,16,17,18,19] and a cross-sectional survey completed by hemodialysis patients (*n* = 2) [20]. The years of publication ranged from 2015 to 2021 and the countries of publication included the USA (*n* = 3) [15,16,17], Australia [9], Korea [19], Saudi Arabia [20], India [10] and Jordan [18]. All studies utilized patient data records and/or perspectives for data collection, excluding the online survey [9] which recorded pharmacists’ perceptions only. Across the eight studies there were 4731 participants, with sample size ranging from 34 to 2199 participants.

Effectiveness of the pharmacist interventions were measured both qualitatively and quantitatively. Quantitative methods collected biometric markers, including blood pressure and blood serology for drug concentrations as indicators of compliance [17,18]. One study reported on drug-related discrepancies identified upon discharge from hospital [19] while another utilized the Pharmacy Quality Alliance Proportion of Days Covered calculation to report adherence as an indicator of compliance [16].

### 3.3. Barriers to Adherence

Barriers to medication adherence in patients with CKD were reported [9,15,16,17], with two overarching themes, literacy and health access, emerging. Sub-themes have been linked to these two main themes in the following way: literacy (lifestyle, regimen complexity, confusion) and health access (cost, logistics). Many patients undergoing dialysis forget to take their prescribed medicines, change their mediation regimen according to their lifestyles and have a very limited understanding or are confused about their medications and/or their CKD condition [15]. Complex medication regimens, compounded with unnecessary medications, with limited family and social support were also reported barriers [9,19]. Yeung et al. utilized the modified Pharmacy Quality Alliance Proportion of Days Covered (PDC) model to measure medication adherence before and after exposure to low health literacy educational flashcards and videos [16]. Health literacy was identified as being an important determinant of medication adherence [16,17]. Age and adherence were significantly correlated, with nonadherence significantly higher in patient groups younger than 50 years of age (*p* < 0.05) [10]. 

Health access was the second barrier that emerged from the review. The financial burden, including the cost of treatments and low family income, as well as the logistical challenges of accessing medicines, were identified as being major determinants of patient nonadherence [10,15]. Nonadherence to medication regimens was prevalent with particularly low adherence rates for antihypertensive medications such as angiotensin-2 receptor blockers, angiotensin converting enzyme inhibitors and dihydropyridine calcium channel blockers [15]. 

### 3.4. Interventions

Common pharmacist interventions discussed in the included studies were medication reviews (*n* = 7) [9,10,15,16,17,18,19], including the revision of doses and administration, motivational counselling (*n* = 3) [9,10,19], tele-health medication counselling (*n* = 2) [10,18], patient education about their condition, concomitant conditions and medicines (*n* = 7) [9,10,15,16,17,18,19], hospital discharge care transition plans (*n* = 1) [19] and collaborative healthcare provision alongside doctors (*n* = 3) [17,18]. 

The role of multimodal disease and medication education in pharmacist-led interventions for the improvement of medication adherence in patients with CKD was reported [16,20]. Studies utilized smart-phone activated quick-response (QR) codes linked to educational videos and flashcards to communicate information to patients. Patients in an intervention group who watched educational videos regarding their medicines and disease states increased medication adherence to their prescribed medication regimes by 29% more than those in the control group, who did not view the educational materials (*p* < 0.001) [16,20]. In patient groups with low literacy skills, the use of QR-coded educational videos and images, which capitalized on using images and spoken phrases using simple terminology, significantly improved patient medication adherence (*p* < 0.01) [16,20]. Of note in the study by Yeung et al., the intervention period was 180 days, which is an extensive intervention at targeting knowledge and resulted in improvement in adherence according to dispensing data compared to the controls (71% vs. 44%; *p* = 0.0069) [16]. Similarly, in the study by Qudah et al., which targeted both knowledge and behavior by monitoring patient home blood pressure monitoring engagement, blood pressure reductions and adherence to medication and attendance at dialysis sessions, those in the intervention group who received weekly engagement with their pharmacist or physician over a 15 week period had positive improvements [18]. Forty-six percent of patients in the intervention arm achieved BP target compared to only 14.3% of patients in the control arm (*p* = 0.02) [18]. Average decline in weekly mean home systolic blood pressure was 10.9 ± 17.7 mmHg in the intervention arm (*p* = 0.004), while weekly mean home systolic blood pressure increased by 3.5 ± 18.4 mmHg in the control arm (*p* = 0.396) [18]. However, there was no significant reduction in weekly home diastolic blood pressure, dialysis blood pressure readings or interdialytic weight gain in either arm of the study [18]. 

Interprofessional collaborative care between pharmacists and physicians was shown to have a positive effect on patient adherence through reducing adverse events and optimizing therapeutic outcomes [17,18]. The inclusion of a pharmacist in a team supporting patients with CKD improved team/physician adherence to primary care guidelines [17]. 

Song et al. identified that pharmacist-led medication reviews at discharge significantly reduced the number of drug-related errors with a large degree of statistical significance (*p* < 0.001) [19]. Data collected to record the number of drug-related errors at discharge, whether the patient requires an appointment with a doctor regarding their CKD therapy within three months of discharge and whether patients present to the emergency department within six months of discharge secondary to problems with their CKD medications supported these findings [15,19]. Song et al. also reported that medication adherence scores were higher in the intervention group, but not significantly so [19].

### 3.5. Pharmacist and Patient Perceptions

Pharmacist-perceived importance of pharmacist-led interventions and their efficacy was discussed in three of the studies [9,15,20]. Pharmacists identified that they perceive themselves to be an extremely effective and yet a highly under-utilized asset for improving patient medication adherence [9]. This was attributed to a lack of time provided for applying interventions for improving medication adherence and limited pharmacists employed to fulfil the roles [9]. This was supported by the patient perceptions, where it was reported that 77.5% and 92% of patients found pharmacist interventions to be an extremely satisfying and effective means of improving adherence and would recommend the service to others [17,20]. 

Patients also reported their preferences for multi-modal delivery of pharmacist interventions [10]. Patient counselling supplemented with patient information leaflets was the most preferred method (68%), followed by patient counselling alone (20%) and patient leaflets alone (12%) [10]. A preference for medication counselling to occur during dialysis (42%) was also reported, with agreement that the provision of educational materials pertaining to medication during dialysis would be helpful (77.5%). Less than half of the participants (37.7%) believed that their physician should be performing the medication counselling role [10].

### 3.6. Financial Implications

Daifi et al. discussed the financial implications of clinical pharmacist interventions in hemodialysis (HD) settings [15]. Interventions including medication reconciliations, medication reviews and patient counselling to improve medication adherence were identified as helping reduce the financial burdens associated with further unnecessary doctors’ visits, avoiding emergency department presentations and avoiding hospital admissions secondary to drug-related discrepancies [15]. In this study, pharmacists completed a medication reconciliation and medication review (at least fou occasions) of hemodialysis patients over a 12 month period. These savings accumulated to a total estimated saving of AUD 447,355 per annum [15]. 

### 3.7. Overall Outcomes

All studies identified that CKD patient medication adherence increased when patients are provided with pharmacist-led services, including medication reconciliations, medicine reviews and counselling on their medications and condition. The studies used measurement tools to determine adherence including the Modified Morisky Scale (MMS) and less rigorous methods including patient reports and pharmacist perspectives. In the study by Yeung et al., the medication adherence score measured by MMS at a patients first outpatient visit after discharge was higher in the intervention group (5.2 ± 1.0) than in the control group (4.9 ± 1.3), but not significantly so (*p* = 0.205) [19]. These pharmacist services have been found to be a generally under-utilized and under-allocated resources [9,10,15,19,20]. The pharmacist was also reported to have an integral role in interdisciplinary healthcare provision with doctors to further enhance patient medication adherence [18]. The success of pharmacist-led education also increased greatly when they are accompanied by well-developed multimodal educational aids [16,17].

**Table 1 pharmacy-11-00185-t001:** Data extraction table for included papers (N = 8).

Author, Year, Country and Aim	Study Design: Participants:	Methodology (and Analysis)	Study Findings
Al-Abdelmuhsin et al., 2020, Saudi Arabia [20] To assess the satisfaction of patients undergoing hemodialysis regarding counselling services provided by pharmacists.	**Study design:** cross-sectional survey—self reporting questionnaire. **Participants:** *n* = 138 patients. Age: 51–75. On dialysis for 1–5 years with comorbidities. **Setting:** outpatient pharmacy service at hospital.	**Methodology:** 224 hemodialysis (HD) patients in the KAMC-Central Region (KAMC-CR) completed a satisfaction survey recording demographic data, HD duration, preferred counselling time, detailed medications list and preferred health professional for medication counselling. **Analysis:** descriptive data were summarized and analyzed using chi-square tests and Fisher’s exact tests (*p* < 0.05).	**Barriers:** ☒ **Interventions:** ☒ **Perceptions:** ☒ **Financial Implications:** ☒ **Outcomes:** ☒ pharmacists play an integral role in providing patients with knowledge regarding their treatment and thus improving their relationship with their often-extensive therapies.
Chandrasekhar et al., 2018, India [10] To evaluate the effect that various interventional methods have on medication adherence behaviors of patients with CKD.	**Study design:** prospective interventional study—modified Morisky eight item questionnaire and self-reporting questionnaire. **Participants:** *n* = 163 patients. Age (years): ≥76 (9%), 61–75 (40%), 46–60 (28%), 31–45 (16%), <30 (7%). 48% in stage 5 of CKD, 32% in stage 4 of CKD and remaining 20% were in stage 3 of CKD. None of the study subjects were in stage 1 or 2 of CKD. **Setting:** outpatient pharmacy service at hospital.	**Methodology:** the Modified Morisky 8 item Questionnaire (MMQS-8) was used to characterize patient adherence. Interventions including patient counselling, patient information leaflets and tele-health consults were provided to patients based on their adherence scores. Post-interventional patient adherence was recorded for comparison. **Analysis:** Chi square tests, independent *t* tests, ANOVA techniques and paired *t* tests were all used to analyze the statistical significance in adherence score changes and differences between participant groups.	**Barriers:** ☒ **Interventions:** ☒ **Perceptions:** ☒ **Financial Implications:** ☒ **Outcomes:** ☒ periodic counselling by clinical pharmacists of patients with CKD improves medication adherence through improving comprehension and removing misconceptions regarding the disease and therapy. Statistically significant association exist between medication adherence before and after intervention (*p* <0.001).
Cooney et al., 2015, USA [17] To evaluate the effect of a pharmacist-based quality improvement program on patient outcomes and adherence to CKD guidelines in a primary care setting.	**Study design:** pragmatic, randomized, controlled trial—direct measure of parathyroid hormone and blood pressure. **Participants:** *n* = 2199 patients. Age: mean—75.6 years. 76%—stage 1, 2, 3, 18%—stage 4, 6%—stage 4–5. **Setting:** outpatient pharmacy service at medical center.	**Methodology:** patients were selected from community based outpatient clinics (moderate to severe CKD). The intervention arm included pharmacist medicine reviews and counselling, pharmacist collaboration with physicians and the provision of information booklets to patients. Clinical outcomes indicated quality of life and all-cause mortality recorded as indicators. **Analysis:** for categorical outcomes, control and interventional arms were compared using Chi-squared tests. Impact of interventions evaluated using *t* tests.	**Barriers:** ☒ **Interventions:** ☒ **Perceptions:** ☒ **Financial Implications:** ☒ **Outcomes:** ☒ pharmacist-led medication reviews, counselling, collaboration with physicians and patient information resources improved medication adherence in patients with CKD.
Daifi et al., 2021, USA [15] To evaluate the impact of a clinical pharmacist in a single hemodialysis (HD) facility on patient medication outcomes and compliance.	**Study design:** retrospective observational descriptive study—clinician assessment of drug-related problems. **Participants:** *n* = 2000 patients. Age: mean 63 (26–92). All patients on hemodialysis. **Setting:** outpatient hemodialysis facility.	**Methodology:** HD patients clinical notes made by pharmacists during daily medication reconciliations and medication reviews were recorded in patient electronic medical records. **Analysis:** patterns in medication-related problems (MRP), drug classes associated with medication-related problems and overall outcomes were identified and considered.	**Barriers:** ☒ **Interventions:** ☒ **Perceptions:** ☒ **Financial Implications:** ☒ **Outcomes:** ☒ pharmacist interventions are an effective means of addressing medication-related programs for patients with CKD, resulting in improved medication adherence, patient outcomes and alleviating financial burden on the healthcare system.
Ghimire et al., 2018, Australia [9] To measure Australian renal-specialized pharmacists’ perceptions, current practices and barriers to assessing adherence in dialysis patients.	**Study design:** cross-sectional online survey—pharmacist survey. **Participants:** *n* = 41 renal-specialized pharmacists. **Setting:** Public and private dialysis units.	**Methodology:** survey questions (10 point Likert scale) demographics, medication adherence, contributors to nonadherence, perceived effectiveness of methods to identify adherence, barriers to assessing adherence and pharmacists’ confidence in assessing adherence. **Analysis:** descriptive analysis and analysis using Dunn–Bonferroni test to identify statistically significant differences between groups with a *p*-value threshold of <0.05.	**Barriers:** ☒ **Interventions:** ☒ **Perceptions:** ☒ **Financial Implications:** ☒ **Outcomes:** ☒ importance of a designated renal pharmacist in clinical settings for the assessment and counselling of dialysis patient medication adherence highlighted.
Qudah et al., 2016, Jordan [18] To evaluate the applicability of a physician and pharmacist collaborative model in the management of blood pressure in HD patients.	**Study design:** randomized controlled, block design clinical study—direct measure of blood pressure. **Participants:** *n* = 56 patients. Age: mean 52 years ± 18. Hemodialysis patients. **Setting:** inpatient hospital.	**Methodology:** in the interventional arm, pharmacists reviewed patient blood pressure (BP) readings and provided advise to physicians to optimize pharmacotherapy. Patients were also provided with educational materials and counselled by the pharmacist. Patient BP was used as the indicator for therapy efficacy. **Analysis:** continuous data were reported as mean ± standard deviation for normally distributed data and independent t-tests were used to detect differences. Categorical data were expressed as frequencies and percentages and compared using chi-squared tests. Significance was set to 0.05 and the confidence interval at 95%.	**Barriers:** ☒ **Interventions:** ☒ **Perceptions:** ☒ **Financial Implications:** ☒ **Outcomes:** ☒ implementation of collaborative pharmacist-physician interventions significantly improved medication adherence and patient outcomes.
Song et al., 2021, Korea [19] To analyze the effectiveness of clinical pharmacist services on drug-related problems and patient outcomes in in patients with chronic kidney disease (CKD).	**Study design:** prospective, randomized, parallel, controlled clinical trial—clinician assessment of drug related problems. **Participants:** *n* = 100 patients. Age: mean 51 years ± 17. Hemodialysis patients. **Setting:** inpatient hospital.	**Methodology:** The intervention group received pharmacist-led medication reconciliation, medication evaluation and management reviews and discharge pharmaceutical care transition (dPCT) services. Outcomes were measured using the number of drug-related problems (DRP) at discharge. **Analysis:** Categorical variables are presented as numbers and percentages; continuous parametric data is presented as mean values and standard deviations and nonparametric continuous data are represented as medians and interquartile ranges.	**Barriers:** ☒ **Interventions:** ☒ **Perceptions:** ☒ **Financial Implications:** ☒ **Outcomes:** ☒ hospital Pharmacist interventions resulted in a significant reduction in drug-related discrepancies (*p* < 0.001) in patients with CKD.
Yeung et al., 2017, USA [16] To design and investigate a pharmacist-run intervention using low health literacy flashcards and smartphone-activated quick response (QR) barcoded educational flashcard video to increase medication adherence and disease state understanding.	**Study design:** prospective, matched, quasi-experimental design—outcome measure—medication possession ratio. **Participants:** *n* = 34 participants. Age: mean 52 years ± 8. Patients with diabetes, heart failure and/or hypertension. **Setting:** community-oriented outpatient clinic.	**Methodology:** patients medication adherence was measured using the modified Pharmacy Quality Alliance Proportion of Days Covered (PDC) model. Interventional group patients were given targeted low health literacy educational resources. These included quick-response barcoded educational flashcards and videos regarding relevant medications. Post-interventional medication was then recorded for comparison. **Analysis:** descriptive statistics were used to describe demographic data and baseline patient characteristics. The Wilcoxon signed ranked test was used to compare the differences between the initial and post-interventional PDC results.	**Barriers:** ☒ **Interventions:** ☒ **Perceptions:** ☒ **Financial Implications:** ☒ **Outcomes:** ☒ pharmacist-led use of flashcards and QR-coded prescription bottles is an innovative and effective means of improving medication adherence and disease state understanding in low-health literacy patient populations.

☒ Indicates this topic was described in the article.

## 4. Discussion

### 4.1. Summary of Evidence

The purpose of this review was to describe the available evidence on the role of pharmacists in providing interventions to improve medication adherence of patients with CKD. This review has identified a limited knowledge base, and this needs to be remembered when considering the broader application of the results. However, it highlights important knowledge gaps in this research area for future investigation. Pharmacists can play an integral role in mitigating the factors which limit patient adherence, but there is little detail in the findings to clearly outline how pharmacist-led interventions are utilized and how they can be improved. The eight studies included in this scoping review discuss the barriers to patient medication adherence, pharmacist and patient perceptions of pharmacist-led interventions, and the efficacy of the current interventions for patients with CKD [9,10,15,16,17,18,19,20]. The ‘donut model’, based on the World Health Organizations’ five determinants of medication adherence, has been used to guide the discussion [6], including condition- and medication-related factors, the healthcare system and healthcare professional-related factors, patient-related factors and socioeconomic factors [21]. 

### 4.2. Condition- and Medication-Related Factors

CKD is a complex condition that often co-exists with other health conditions, requiring complex medication regimes. Characteristic of CKD, polypharmacy can result in large pill burdens and patient confusion regarding administration instructions [17]. Large numbers of prescribed medicines also place patients at increased risk of experiencing adverse events or contraindications with foods and prescribed or complementary medicines [22]. Pharmacist-led counselling on patient conditions can provide relevant information to help patients adequately understand their conditions [16]. The studies included in this review identified that providing patients with a clear understanding of their conditions improved medication adherence [9,10,19,20]. This is likely due to patients gaining a basic understanding of not only the role of the renal system and the consequences of renal impairment, but also how pharmacological management can greatly improve outcomes and quality of life [1,2,22].

Pharmacists play a role in mitigating medication-related factors that would reduce medication adherence in patients with CKD [15,19,20]. Medication reviews and patient counselling were interventions used by pharmacists to identify potential barriers to treatment success and patient adherence to prescribed regimes [23]. Pharmacist-led medication reconciliation, medicine reviews and hospital discharge care transition plans all resulted in simplified patient regimes and decreased the number of medications prescribed at discharge from hospital [19]. These factors will all have contributed to the significant increase in post-interventional medication adherence seen amongst cohorts in interventional compared to control groups [10,15,17,20]. Patient education during pharmacist counselling regarding medications including the indications of prescribed drugs, general mechanism of action of medications and possible common adverse effects all further improved patient adherence [15,19,20]. Studies found that providing patients with adequate information to understand their conditions and therapies improved their perception of their therapies and increased their medication adherence [15,19,20].

### 4.3. Healthcare System and Healthcare Professional-Related Factors

Pharmacist interventions are limited by the healthcare systems in which they practice [9,10,15,17]. Many pharmacists recognize their interventional services are having a positive effect on patient adherence, but are restricted by the limited of time and resources which are allocated to the services [9]. The lack of pharmacist roles designated specifically to medication reconciliation, medicine reviews and patient counselling and education in renal care and dialysis settings limits the time which can be spent with each patient [9]. This means that pharmacists are rarely granted the time it would take to properly ensure that patients have an adequate understanding of what medicines they are prescribed, why they are prescribed them and to ensure patients follow the dosage instructions [9]. Providing patients with this knowledge has been identified as significantly increasing medication adherence amongst patients with chronic kidney disease [10,15,17,19,20]. This can be attributed to simplifying and demystifying the aspects of their treatment and condition, which are often quite confusing and not well understood [10,15,17,19,20]. Therefore, by limiting pharmacists’ opportunities and abilities to provide these services limits optimal medication adherence in patients with CKD [9].

### 4.4. Patient-Related Factors

Patients’ preferred language and level of education and literacy were identified as strong determinants of medication adherence [16,17]. A positive correlation between patient engagement in health education resources that included diagrammatic flashcards and educational videos and medication adherence has been reported [16]. Patients were able to gain understanding of their medications and conditions using these resources, which capitalized on images and the use of spoken language using simple vocabulary and were found to be effective regardless of the patients’ prior level of health literacy [16]. Pharmacists possess the knowledge and skills to utilize these educational resources to supplement their communication during patient counselling to improve information retention, comprehension and consequential adherence to prescribed medicines [16]. These resources also proved to be applicable to patients who are less open to counselling by pharmacists where they can access information on their personal devices through scanning a quick-response (QR) code [16]. The development of these tools could provide an opportunity to break down the language barrier of patients, where resources can be translated into various languages to engage patients from different language backgrounds. The use of images and simple vocabulary also allows pharmacists to communicate clearly to patients, without the barriers of the potentially intimidating nature of technical jargon [24]. From the studies identified, the impact of behavioral-only or mixed education and behavioral interventions for CKD have not been clearly reported and is thus unknown [25].

### 4.5. Socioeconomic Factors

Pharmacist-led patient counselling, medication reconciliation, medicine reviews and hospital discharge care transition plans resulted in optimized drug therapy, reducing the number of unnecessary visits to doctors, emergency department presentations and hospital admissions [19]. This reduces the financial burden placed on patients with CKD. Reducing the financial burden of accessing their prescribed medicines, as well as reducing the need for further medical interventions secondary to suboptimal regimes, all increases medication adherence amongst patients with CKD [19].

### 4.6. Strength and Limitations 

This is the first scoping review to synthesize literature related to the pharmacist’s role in medication adherence in CKD. The review focusses on peer-reviewed published papers, which may have resulted in some papers in the gray literature, opinion articles and editorial reviews being omitted. The study is limited to articles indexed on the databases searched and those in English and therefore it is possible that some published literature was not sourced. Publication bias, variety in study design, not assessing study quality and the limited articles reviewed also present a limitation to this review. Diabetic nephropathy is a common concomitant condition which often leads to the development and worsening of CKD [26]. The exclusion of literature which referred to diabetes but not to chronic kidney disease could have removed information which was relevant from the patient’s perspective. These limitations warrant consideration in the planning of future reviews.

### 4.7. Further Research and Knowledge Gaps

The broad role of the pharmacist in caring for patients with CKD is reported in the literature, however the actual impact on pharmacist interventions for patients with CKD is limited and at most, after a systematic review, can be reported to potentially have a positive impact on patient outcomes [26]. Pharmacists working within multidisciplinary renal care teams can provide multidimensional interventions and contribute to identification of drug-related problems, positive improvement in clinical outcomes, improved health-related quality of life and patient satisfaction and cost saving through pharmaceutical care provision is reported, however, evidence is of low quality and insufficient volume [27,28]. Thus it is important to realize that more high-quality research is warranted [27,28]. Clear improvement of adherence to medication among patients with CKD as a direct result of pharmacist interventions is non-existent and is needed to shape future pharmacy interventions [28].

Examining the effectiveness of the interventions mapped and the impact on health outcomes for patients with CKD of differing levels of severity would be a useful future direction for further studies. In addition, a more systematic review may identify other opportunities for pharmacists to support medication adherence in kidney health. Further research may improve patient education through the development of educational tools such as videos and images describing medications and the CKD condition [16]. These resources should capitalize on visual engagement and communication with words being limited to exclusively where necessary and to only use simple vocabulary [16,29,30]. All educational tools should be translated into applicable languages to ensure engagement and effective communication to all patients [16,30]. Integrating pharmacists into renal care facilities with the dedicated role of providing medication reconciliation, medicine reviews and counselling to patients is recommended and requires further research [9,10,15,17,19,20]. Specifically, future research to explore the impact of behavioral-only or mixed education and behavioral interventions for CKD would be valuable as it has been reported that educational-only interventions may lack impact [25]. There is significant value in highlighting the improvement of clinical outcomes as a result of intervention-induced improvement in medication adherence, however, existing literature to demonstrate this is lacking and would be a welcomed future research topic.

## 5. Conclusions

Pharmacist interventions have been reported to be effective in improving medication adherence in patients with CKD. Optimizing and simplifying medication regimes and empowering patients with adequate knowledge regarding their condition, comorbidities and medications were all identified as effective goals for improving medication adherence. Important interventions include medication reconciliation, medicine reviews, patient counselling and patient education, however, pharmacy-led services were under-utilized. To improve effectiveness of medication counselling, education tools should capitalize on visual learning and simple vocabulary to enable easy communication with patients regardless of level of health literacy or language. This review provides evidence to support integration of pharmacists dedicated to medication adherence in renal care facilities, with future research needed to evaluate the effectiveness and outcome of such initiatives. 

## Figures and Tables

**Figure 1 pharmacy-11-00185-f001:**
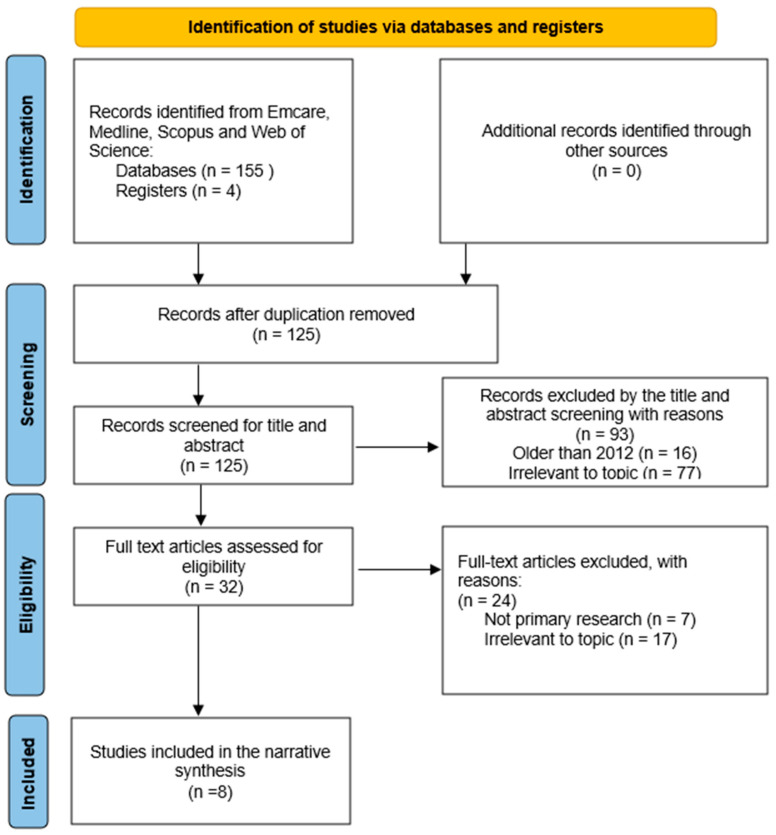
Flowchart of scoping review [14].

## Data Availability

No new data were created or analyzed in this study. Data sharing is not applicable to this article.

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
