# Peer review of "Pharmacist-Led Interventions for Medication Adherence in Patients with Chronic Kidney Disease: A Scoping Review"

_pharmacy, 2023, doi:10.3390/pharmacy11060185_

Round 1

Reviewer 1 Report

Comments and Suggestions for Authors

Thank you for the opportunity to review this promising manuscript, which is a scoping review of the pharmacists' role in medication adherence for patients with CKD. I have attached a copy with comments and suggested edits in track changes below for the authors to address, especially:

1) Wording used to describe the gap/purpose of the scoping review in the introduction

2) Organization and detail in the methods and results, such as the narrative matching Figure 1

3) Additions to the Future research section in the Discussion

Comments on the Quality of English Language

Minor editing of English language required

Author Response

Thank you for you effort and time in reviewing this article. 

Revisions have been made to address all of your recommendations and comments in the article.

1)            Wording used to describe the gap/purpose of the scoping review in the introduction

Thank you – this has also been revised with your suggestion included.

2)            Organization and detail in the methods and results, such as the narrative matching Figure 1

Thank you – the figure and the narrative associated has been revised and corrected.

3)            Additions to the Future research section in the Discussion

Thank you – these two suggestions are valuable and have been added to the section.

In addition, all of your suggested  within text editing has also been included in the revised version

Reviewer 2 Report

Comments and Suggestions for Authors

While I commend the authors for tackling an important health topic and role of pharmacists, it seems to me that the work needs substantial work.

Methods:

-          A scoping review typically follows a multistage process of study identification, selection, data charting and summarising and presenting of findings. It seems to me the method section is a bit too concise without enough clarity on the study selection process or data charting and presentation procedures.

-          More clarification would be great on the characteristics of patients targeted. Does this include people with kidney failure? For example, those on dialysis or had transplant. This need to be clearly defined upfront.

-          Why restricting the search to the past 10 years? Since this is a scoping review, wouldn’t it be better to capture every available evidence?

-          Also, the search was conducted in Nov 2022, did the authors check if there is any emerging evidence since?

-          I am surprised with the small number of articles you found during your search.

-          I believe the side arrow in the PRISMA diagram showing the articles excluded due to not fulfilling the eligibility criteria is misplaced (this is where 81 articles are excluded). It should go one level up. It should be 125-81 = 44.

Results

-          Change the study design for the survey to cross-sectional (i.e., ref #20).

-          Barriers: The two themes identified, i.e., literacy and health access, seem too limited in my opinion. I am sure the included studies have identified several other barriers unless you’re specifically referring to those that are potentially modifiable. This makes it even more confusing as you went on to discuss other barriers like regimen complexity, age, etc.

-          I am not sure findings reported in paragraph 4 under 3.4 is well aligned with what the authors are trying to address. Did the Song et al study report anything related to adherence?

-          The table needs to be restructured and presented better. It’s better to add some more columns than providing lots of information in limited spaces. It’s difficult to easily understand it in its current shape.

-          Very little is known about the interventions and more importantly the patients targeted through the included interventions.

-          You mentioned the interventions are effective in improving adherence, but I couldn’t see what the metrics are to show the effectiveness. How was adherence measured? Are these objective or subjective in nature? What changes are indicators for improvement in adherence?

-          Overall, the results part needs significant work to spell out the main findings when it comes to the effect of the implemented interventions on medication adherence.

Discussion

-          While the discussion is well detailed, this needs to be aligned better with the findings the authors reported.

-          Due to the limited information on the patients targeted or included in the studies, I am not sure to what extent most of the discussion points are applicable to patients at different stages of CKD.

-          It is important to highlight if the included studies reported any improvement in clinical outcomes as a result of intervention-induced improvement in medication adherence. But since I know there are limited outcome studies, it might be important to highlight the limitation of existing literature.

Comments on the Quality of English Language

- Just review the manuscript througout as some of the presentations can be improved, especially around introduction and results.

Author Response

Thank you for your time and effort in reviewing this article.

Methods:

  • A scoping review typically follows a multistage process of study identification, selection, data charting and summarising and presenting of findings. It seems to me the method section is a bit too concise without enough clarity on the study selection process or data charting and presentation procedures.

Thank you – this section has been expanded for improved clarity.

  • More clarification would be great on the characteristics of patients targeted. Does this include people with kidney failure? For example, those on dialysis or had transplant. This need to be clearly defined upfront.

Thank you – the characteristics of the patients has been clarified with more detail. [This included patient with all stages of chronic kidney disease (but excluded acute kidney failure). It also included those undergoing dialysis treatment and those who had received a transplant.]

  • Why restricting the search to the past 10 years? Since this is a scoping review, wouldn’t it be better to capture every available evidence?

This review aimed to report recent and emerging evidence; however, your point is valid and in the future research section we have included commentary about the value of furthering this topic area to include a systematic review.

  • Also, the search was conducted in Nov 2022, did the authors check if there is any emerging evidence since?

As the search was conducted in Nov 2022, the results have been reported from this data.

  • I am surprised with the small number of articles you found during your search.

Agree, thus we have made a recommend for a systematic review in future and plan to work toward this going forward.

  • I believe the side arrow in the PRISMA diagram showing the articles excluded due to not fulfilling the eligibility criteria is misplaced (this is where 81 articles are excluded). It should go one level up. It should be 125-81 = 44.

Thank you this has been corrected.

Results:

  • Change the study design for the survey to cross-sectional (i.e., ref #20).

Thank you – corrected.

  • Barriers: The two themes identified, i.e., literacy and health access, seem too limited in my opinion. I am sure the included studies have identified several other barriers unless you’re specifically referring to those that are potentially modifiable. This makes it even more confusing as you went on to discuss other barriers like regimen complexity, age, etc.

The barriers have been grouped into the two over-arching themes of literacy and health access. You are right, other lower-level themes such as (cost linked to access) (complexity linked to literacy) have been linked to one of the two main themes.  The following has been added for clarity [Sub-themes have been linked to these two main themes in the following way; literacy (lifestyle, regimen complexity, confusion) and health access (cost, logistics).]

  • I am not sure findings reported in paragraph 4 under 3.4 is well aligned with what the authors are trying to address. Did the Song et al study report anything related to adherence?

Thank you – for improved alignment I have also included [Song et al. also reported that medication adherence scores were higher in the intervention group, but not significantly so. [19]]

  • The table needs to be restructured and presented better. It’s better to add some more columns than providing lots of information in limited spaces. It’s difficult to easily understand it in its current shape.

Thank you – happy to extend into a landscape orientation and spread the data into more columns with the final editing process.

  • Very little is known about the interventions and more importantly the patients targeted through the included interventions.

Thank you – the results table provides detail including haemodialysis patients and CKD patients and it is hoped that interested readers seek the full articles to learn more.

  • You mentioned the interventions are effective in improving adherence, but I couldn’t see what the metrics are to show the effectiveness. How was adherence measured? Are these objective or subjective in nature? What changes are indicators for improvement in adherence?

Thank you – this has been added. The studies used measurement tools to determine adherence including the Modified Morisky Scale (MMS) and less rigorous methods including patient reports, and pharmacist perspectives.

  • Overall, the results part needs significant work to spell out the main findings when it comes to the effect of the implemented interventions on medication adherence.

Thank you – the summary of the outcomes is included in the results table and discussed through the narrative.

Discussion

  • While the discussion is well detailed, this needs to be aligned better with the findings the authors reported.

Thank you – edits have been made throughout the discussion.

  • Due to the limited information on the patients targeted or included in the studies, I am not sure to what extent most of the discussion points are applicable to patients at different stages of CKD.

Thank you – this is an interesting point, and this scoping review can be applied broadly to CKD patients, however an interesting future direction would be to investigate whether adherence interventions have different outcomes on patients depending on their degree of kidney impairment. The following has been added [Examining the effective of the interventions mapped and the impact on health outcomes for patients with CKD of differing levels of severity would be a useful future direction for further studies.]

  • It is important to highlight if the included studies reported any improvement in clinical outcomes because of intervention-induced improvement in medication adherence. But since I know there are limited outcome studies, it might be important to highlight the limitation of existing literature.

Thank you – this point has been included in the future research section. [ There is significant value in highlighting the improvement of clinical outcomes because of intervention-induced improvement in medication adherence, however existing literature to demonstrate this is lacking and would a welcomed future research topic.]

Reviewer 3 Report

Comments and Suggestions for Authors

Please see the word file attached.

Author Response

Thank you for your time and effort in reviewing this article.

Please find revisions below.

  • I suggest minor revisions to the manuscript before publication, mainly by adding some more information to the result and discussion chapters. The evidence base identified is limited (8 studies included) with various study designs with varying quality of evidence. Therefore, I think it is appropriate to point out to the reader that the knowledge base this review concludes from is very limited. However, the result highlights the knowledge gap in this research field, which should be pointed

  • Thank you – the following has been added to the beginning of the discussion. [This review has identified a limited knowledge base, and this needs to be remembered when considering the broader application of the results. However, it highlights important knowledge gaps in this research area for future investigation.]

Abstract:

  • The aim of study differs from the one presented in the Please, be consistent.
  • Thank you – the aim in the abstract is a condensed version of the aim at 1.1. The introduction has been revised.

Introduction:

  • From my perspective, the introduction is well written and provides a general background to the medication adherence scenario among CKD patients, and the potential role of pharmacists. However, I was curious to read about pharmacist-led interventions targeting this patient population in general. For example, it could be relevant to present results from the following papers:

Salgado TM, Moles R, Benrimoj SI, Fernandez-Llimos F. Pharmacists' interventions in the management of patients with chronic kidney disease: a systematic review. Nephrol Dial Transplant. 2012 Jan;27(1):276-92. doi: 10.1093/ndt/gfr287.

Al Raiisi F, Stewart D, Fernandez-Llimos F, Salgado TM, Mohamed MF, Cunningham S. Clinical pharmacy practice in the care of Chronic Kidney Disease patients: a systematic review. Int J Clin Pharm. 2019 Jun;41(3):630-666. doi: 10.1007/s11096-019-00816-4.

-Stroschein M. The Effectiveness of Pharmacist Interventions in the Management of Patient with Renal Failure: A Systematic Review and Meta-Analysis. Int J Environ Res Public Health. 2022 Sep 6;19(18):11170. doi: 10.3390/ijerph191811170.

  • These studies might also highlight the need to perform this scoping review to elucidate on adherence interventions and outcomes. Moreover, such studies are interesting in discussing your results; Pharmacist-led interventions among CKD patients targeting adherence outcomes, are there relevant differences to interventions targeting other outcomes? What can we learn from these interventions compared to other interventions in management of medication outcomes in CKD?
  • Thank you for providing these articles – they will be very valuable for a future systematic review for this topic areas that I am planning. As they are not directly related to the specific aim of this scoping review, I have not included them in the introduction. I have however included them in the discussion.
  • The aim of the introduction is different from the aim in the abstract, where highlight the impact

, is left out in the latter. Characterization of interventions vs characterization of literature; the latter being more comprehensive? Consider revising to be consistent.

  • Thank you the aims have been aligned.

Method:

  • Search strategy: the reader will need the comprehensive list of search/key words and how these were combined, to be able to replicate your search. Please add more information for

  • Thank you this detail has been added.

  • Extraction of results and data synthesis: Please add more information about data extraction, analysis, and synthesis process. Which data categories were predetermined for data extraction? Did you add some categories while analysing? If yes, what kind of data? The subcategorization into themes is this a result of your analysis, or were these predetermined? How did you arrive at this result?
  • Thank you the following has been added [The data charting process was extensive included extracting, analysing and presenting evidence from each study to answer the review question. Data items were initially predefined and were revised and refined through an iterative process with the study authors to form Table 1.]

Results:

  • Characteristics of included studies:

There is one study missing when describing countries of publication (India, reference 10).

  • Thank you – I have included it.

I would like to read more about the patients. For example, age distribution of the sample is relevant, since age and adherence are described to be significantly correlated. Grade of CKD is of interest. Numbers of medicines in use

  • Thank you – additional patient details have been added to the results table.

I think it is relevant to describe the setting and context the pharmacist-led interventions are  performed, because there is a variety from hospital to primary care setting. Please consider adding information.

  • Thank you setting has been added to the table.

In this section, measurement of pharmacist interventions effectiveness is described. Is this medication adherence effectiveness measurements? Any self-reported adherence measures   should be included in the description, as these are very commonly used in adherence interventions (for example, MMQS8 is described in Table 1). Moreover, I recommend a clear and comprehensive description and characterisation of medication adherence measures and outcomes, since this is the primary outcome of the scoping review. Furthermore, you should consider categorizing the outcome measurements, see for example Lam WY and Fresco P, 2015 (doi: 10.1155/2015/217047) for inspiration and guidance.

  • Thank you – adherence measures are now included in the table.
  • Interventions: When reading this section, I had several questions that should be considered elucidated: Were the interventions targeting knowledge or behaviour, or both? Were multiple intervention components used, and if yes, what were they? Did interventions include adherence aids, apps or alarms? How long/large were the interventions, for example how many counselling sessions, timeframe of prospective study and retrospective study? Time of follow-up? Intensity (time used, how many pharmacists, and so on) of interventions? Consider adding more information to characterize
  • Thank you – more information has been added to characterize the interventions.
  • Financial implications: Were there any cost-effectiveness outcomes?
  • No other cost-effectiveness outcomes were reported.
  • Overall outcomes: Did interventions target medication adherence as main outcome? This is of importance in understanding impact. Please provide a clear and comprehensive description of adherence outcomes. It is also of interest to present clinical outcomes, as some are presented in Table

Table 1:

Please add all relevant adherence intervention components and outcomes measured. For example, it is not clear to the reader how reference 10, 19, and 20 is linked to adherence. For example, in reference 19, they measure DRPs which were significantly lower in the

intervention group. However, did they also measure adherence? What is the link? Was  nonadherence categorized as a DRP?

Thank you – this detail has been added.

For all references where a significant change in adherence (or other outcome measures) are  described: please report precise observed p values and effect sizes. This will help the reader to understand the meaning and magnitude of results. This is relevant for example to reference 10, 17, 18, 16.

Thank you – more detail has been included. See in Results under Barriers to Adherence for article 10 and Interventions for article 16,17,18

Discussion

  • Please interpret the results according to previous research results presented in the introduction (for example reviews on pharmacist-led interventions targeting CKD patients and other outcomes). All evidence combined: what is the state-of-the-art within pharmacist-led interventions to aid CKD patients? Are there specific areas to be addressed when targeting medication adherence, compared to other outcomes?

Pharmacist-led interventions are aiding CKD patients are  highlighted as follows:

The ‘donut model’, based on the World Health Organizations’ five determinants of medication adherence, has been used to guide the discussion, [6] including condition- and medication- related factors, the healthcare system and healthcare professional-related factors, patient-related factors and socioeconomic factors.[21]

Under each of these deteminants of medication adherence the pharmacist-led interventions are included where appropriate eg

4.2. Condition- and medication- related factors

Pharmacist-led counselling on patient conditions can provide relevant information to help patients adequately understand their conditions.[16] Pharmacist-led medication reconciliation, medicine reviews and hospital discharge care transition plans all resulted in simplified patient regimes and decreased the number of medications prescribed at discharge from hospital.[19]

4.3. Healthcare system and healthcare professional-related factors

This determinant highlights the challenges associated with pharmacist interventions.

Pharmacist interventions are limited by the healthcare systems in which they practice. [9, 10, 15, 17] Many pharmacists recognise their interventional services are having a positive effect on patient adherence, but are restricted by the limited of time and resources which are allocated to the services. [9]

4.4. Patient-related factors

Pharmacists possess the knowledge and skills to utilize these educational resources to supplement their communication during patient counselling to improve information retention, comprehension and consequential adherence to prescribed medicines.[16] These resources also proved to be applicable to patients who are less open to counselling by pharmacists where they can access information on their personal devices through scanning a quick-response (QR) code. [16]

4.5. Socioeconomic factors

Pharmacist-led patient counselling, medication reconciliation, medicine reviews and hospital discharge care transition plans resulted in optimised drug therapy, reducing the number of unnecessary visits to doctors, emergency department presentations and hospital admissions.[19]

  • Please remind the reader of the limited knowledge base this review is discussing and concluding
  • This review has identified a limited knowledge base and this needs to be remembered when considering the more broad application of the results.
  • Strength and limitations: The study limitations must be extended, for example publication bias, limited number of evidence, not assessing study quality. The variety in study designs might also be relevant to
  • Thank you – this has been added [Publication bias, variety in study design, not assessing study quality and the limited articles reviewed also present a limitation to this review.]

  • Further research: The focus on further research on educational tools is somewhat controversial to other adherence intervention reviews and meta-analysis, that recommend multidimensional interventions targeting behaviour over education, or including both (see for example the references in introduction, and Cross et al 2020, DOI: 10.1002/14651858.CD012419.pub2.). Does the result in this scoping review fully support the suggested direction for further research (mainly based on reference 16?)? You could consider highlighting the need for further research including adherence and outcomes, and cost-effectiveness analysis, using high-quality study design?
  • Thank you – the following has been added. Specifically, future research to explore the impact of behavioural only or mixed education and behavioural interventions for CKD would be valuable as it has been reported that educational only interventions may lack impact.[25]

Conclusion

  • I recommend revising the conclusion so that it is fully supported by the main results. For example, pharmacy-led services were under-utilized, under-developed and under-funded: Where can I read about this in results?
  • Thank you – this sentence has been revised to only describe under-utilized.

Round 2

Reviewer 1 Report

Comments and Suggestions for Authors

The second to last sentence preceding Figure 1 should spell out "Sixteen." 

Author Response

Once again, thanks so much for your time. Your correction of 'sixteen' has been made.

Kind regards and Merry Christmas to you.

Reviewer 2 Report

Comments and Suggestions for Authors

Thanks for addressing most of my comments.

I would suggest paraphrasng the title to better reflect the content and also to make sure the use of person-first language. One suggestion I have is as follows but up to the authors to modify this where they see fit: " The effect of pharmacist-led interventions on medication adherence in patients with chronic kidney disease: A scoping review"

Author Response

Thank you for the tittle suggestion - the following has been applied.

"Pharmacist-led interventions for medication adherence in patients with chronic kidney disease: A scoping review"

Once again, thanks for you time on this review.

Kind regards and Merry Christmas
